# Infant Acute Lymphoblastic Leukemia—New Therapeutic Opportunities

**DOI:** 10.3390/ijms25073721

**Published:** 2024-03-27

**Authors:** Marika Kulczycka, Kamila Derlatka, Justyna Tasior, Maja Sygacz, Monika Lejman, Joanna Zawitkowska

**Affiliations:** 1Student Scientific Society of Department of Pediatric Hematology, Oncology and Transplantology, Medical University of Lublin, 20-093 Lublin, Poland; 58224@student.umlub.pl (M.K.); 58530@student.umlub.pl (K.D.); 54238@student.umlub.pl (J.T.); 58267@student.umlub.pl (M.S.); 2Independent Laboratory of Genetic Diagnostics, Medical University of Lublin, 20-093 Lublin, Poland; monikalejman@umlub.pl; 3Department of Pediatric Hematology, Oncology and Transplantology, Medical University of Lublin, 20-093 Lublin, Poland

**Keywords:** acute lymphoblastic leukemia, infant, targeted therapies

## Abstract

Infant acute lymphoblastic leukemia (Infant ALL) is a kind of pediatric ALL, diagnosed in children under 1 year of age and accounts for less than 5% of pediatric ALL. In the infant ALL group, two subtypes can be distinguished: *KMT2A*-rearranged ALL, known as a more difficult to cure form and *KMT2A*- non-rearranged ALL with better survival outcomes. As infants with ALL have lesser treatment outcomes compared to older children, it is pivotal to provide novel treatment approaches. Progress in the development of molecularly targeted therapies and immunotherapy presents exciting opportunities for potential improvement. This comprehensive review synthesizes the current literature on the epidemiology, clinical presentation, molecular genetics, and therapeutic approaches specific to ALL in the infant population.

## 1. Introduction

Infant acute lymphoblastic leukemia (ALL) constitutes a particular subgroup of malignancy referred to children younger than 1 year at the time of diagnosis. This group of patients is assessed as having the worst survival rate and outcome factors determining the prognosis. It constitutes 1% of all pediatric ALL. The majority of cases concern ALL B lineage, while T lineage and mixed phenotype (MPAL) are only a small percentage. In the survival prognosis of the infants’ leukemia, the age of the patients is not insignificant. ALL, contrary to acute myeloblastic leukemia (AML), is characterized by lower infant outcomes compared to older children with similar cytogenetic features. Moreover, infant leukemia is distinguished by aggressive symptoms, high-risk cytogenetic features associated with chemotherapy resistance, and high relapse rates as well as elevated rates of therapy-related toxicities and long-term effects [1,2,3]. In particular, those types with *KMT2A*-rearranged ALL are characterized by hyperleukocytosis, a relatively high incidence of central nervous system (CNS) involvement, an aggressive course with early relapse, and early relapses resulting in poor prognosis [4]. The researchers noticed that different types of rearrangements in the *KMT2A* gene, very high white blood cell count, an age of younger than 6 months, and a poor response to the prednisone prophase were independently associated with inferior outcomes [5]. Moreover, patients with congenital ALL (diagnosed in the first month of life) have been found to have a significantly higher relapse rate [6]. Treatment is based on multidrug chemotherapy consisting of found phases: cytoreductive prophase, induction, consolidation of remission, and maintenance therapy commonly with the subsequent hematopoietic cell transplantation (HCT). The use of HCT in infant leukemia is uncertain due to the fact that only a small minority of *KMT2A*-r patients at high risk of relapse (very young age (<6 months old), very high WBCs (≥300,000/μL), and persistence of minimal residual disease (MRD) appear to benefit from HCT in first remission. Currently, innovative therapies with the modulation of epigenetic factors or signaling pathways and immunotherapy are being tested [3,7]. In this publication, our focus centers on prognostic factors, molecular basis, conventional and novel therapies with its aftereffects, as well as future directions.

## 2. Subtypes of Infant ALL

Based on the molecular background, two subtypes of Infant ALL are distinguished: *KMT2A*-rearranged (*KMT2A*-r) and non-rearranged *KMT2A* (wild type *KMT2A*) [1].

### 2.1. KMT2A-r Subtype

*KMT2A*-r is the most common subtype, present in up to 80% of Infant ALL cases [8] It involves a rearrangement of the histone lysine methyltransferase 2A (*KMT2A*) gene, located on the chromosome 11q23 [9]. Before the HUGO nomenclature change, *KMT2A* was named the mixed lineage leukemia (*MLL*) gene [10]. The domain structure of wild-type *KMT2A* is shown in Figure 1.

*KMT2A*-r is caused by multiple genomic lesions, including internal deletions, tandem duplications, and amplifications, yet predominantly, it is a result of a chromosomal translocation that leads to fusion of the KMT2AN subunit with the C-terminal subunit from one of more than 90 identified partner genes [4,11]. The structure of a fusion protein is presented in Figure 2.

The most commonly identified *KMT2A* fusion partners and frequency of their occurrence in Infant ALL are shown in Table 1 [10,11].

The *KMT2A* gene encodes a protein which is a transcriptional coactivator playing an essential role in the regulation of gene expression during normal hematopoiesis and stem cell differentiation [12]. Regulation of gene transcription involves histone 3 lysine 4 (H3K4) methyltransferase activity, which is controlled by the C-terminal SET domain (Figure 1). The *KMT2A* fusion protein preserves the ability to bind DNA and proteins as the MBM, AT hook, and CxxC domains are retained. Nevertheless, the fusion protein does not include, partially or completely, the regulatory BRD and PHD, repressive RD2 domains, and loses the entire *KMT2AC* subunit with the SET domain (Figure 2), which leads to epigenetic dysregulation and potentially initiates leukemogenesis [9,13].

It has been shown that *KMT2A* rearrangements are acquired in hematopoietic precursors during prenatal development [14]. Maternal exposure to environmental DNA topoisomerase II (DNAt2) inhibitors during pregnancy may increase the risk of *KMT2A*-r leukemia in infants [15].

In approximately 50% of cases of infant *KMT2A*-r ALL, PI3K-RAS mutation can be detected. It is assumed that this may be an independent adverse prognostic factor, however, the exact significance of this mutation is unclear [9,16].

The typical *KMT2A*-r ALL immunophenotype has the CD19- positive, CD10-negative B-lymphoblastic cells, often co-expressed with myeloid-related antigens (for example, CD15, CD33, and CD68 antigens). This suggests a highly immature lymphoid progenitor origin [4,17].

### 2.2. Non-Rearranged KMT2A Subtype

The second Infant ALL subtype, known as non-rearranged *KMT2A*, accounts for approximately 20% of all cases of infant ALL. This variant typically occurs in late infancy, presents a more mature CD10-positive B-cell precursor phenotype, shares similar cytogenetic abnormalities with ALL in older children, and is associated with better outcomes [18,19].

In 20% of cases of infant wild type *KMT2A* ALL, rearrangement of the *PAX5* gene is found and is related with adverse outcomes. On the contrary, *NUTM1* rearrangement, present in around 20–30% of cases, has been shown to relate with a better prognosis [20,21].

The majority, over 90% of the wild-type *KMT2A* ALL cases, occur as CD10-positive B-lymphoblastic leukemia [21] and 4–10% as T-lymphoblastic leukemia [22].

## 3. The Past and the Present of Infant ALL Treatment

### 3.1. First Infant-Specific Trials

The therapeutic approach in infant ALL is a major challenge and differs from the childhood ALL treatment [23].

The first clinical trials associated with childhood ALL commenced in the 1950s and have expeditiously shown that infants less than 1 year of age are associated with unfavorable prognosis. Hence, the intensification of a conventional chemotherapy in this age group has been required [24,25]. Consecutive intensification of a conventional chemotherapy in infant ALL patients resulted in drug-related toxicities and led to equalization in survival rate [26].

Thus, new clinical trials on infants based on childhood ALL protocols have begun. Researchers demonstrated that the inferior outcomes are related with features, such as the presence of *KMT2A* rearrangement, hyperleukocytosis at presentation, absence of CD10 antigen, age < 6 months at diagnosis, and poor response to initial prednisone therapy [25].

### 3.2. Current Collaborative Groups

At present, there are three large groups focused on performing clinical trials specific to infant ALL: the Children’s Oncology Group (COG), the Japanese Pediatric Leukemia/Lymphoma Study Group (JPLSG), and the Interfant Study Group [25].

As early relapse within 6–9 months of diagnosis is common in infant ALL patients, the COG P9407 trial was designed to deliver shortened, intensified therapy with the elimination of age- and weight-related dose reductions for most chemotherapy agents aiming to improve the event-free survival rate (EFS). P9407 was also modified three times for induction toxicity resulting in three cohorts of therapy. Age ≤ 90 days at diagnosis resulted the most important prognostic factor with the 5-year EFS of 15.5%, compared to 48.5% for those >90 days. Despite the novel approach, EFS remained less than 50% overall in *KMT2A*-r ALL patients [27].

The Japanese Pediatric Leukemia/Lymphoma Study Group conducted two consecutive studies on infant ALL, named MLL96 and MLL98. Patients with *KMT2A*-r ALL were supposed to receive allogeneic HCT (allo-HCT) at their first remission after intense chemotherapy. A high rate of early relapse before the HCT resulted in unsatisfactory outcomes. Nevertheless, infants who received HCT at their first remission reached 3-year post-transplantation EFS of 64.4% [28,29].

Based on these findings, the MLL03 trial was performed. Intensification of the pre-transplantation chemotherapy with high-dose cytarabine and assignment of patients to receive HCT in the early post-remission phase resulted in a high rate (90%) of patients able to undergo HCT in their first remission. However, high induction toxicity, and the relevant number of patients who still relapsed after HCT resulted in a 4-year EFS rate of 43.2% [30]. The latest MLL-10 trial of the JPLSG showed a remarkable improvement. Treatment included intensive chemotherapy and the limited indication for HCT to only those from the high-risk group (according to *KMT2A* status, age, and presence of central nervous system leukemia). The EFS rate for patients with *KMT2A*-r ALL from intermediate and high-risk groups was 66.2% [18].

The Interfant Study Group conducted a crucial multicenter randomized clinical trial named Interfant-99. A total of 482 infant patients aged 0–12 months were enrolled between the years of 1999 and 2005. Consequently, they were classified into standard- and high-risk groups based on their response to 1 week of daily systemic prednisone and one intrathecal dose of methotrexate. The treatment hybrid 2-year protocol was based on a framework of a standard ALL treatment approach, including phases of a four-drug induction with the addition of variable doses of cytarabine and methotrexate, consolidation chemotherapy (MARAM), a reinduction phase (OCTADD), an intensification phase (VIMARAM), and three maintenance phases. High-risk patients could also receive, if a donor was available, allo-HCT after the reinduction phase. The aims of the study were to assess the outcome of a hybrid treatment schedule in infants with ALL and to assess the efficacy of a late intensification course with high doses of both cytarabine and methotrexate between the reinduction and maintenance phases. The study demonstrated that people treated with the hybrid protocol had higher event-free survival (EFS) than most reported outcomes for the treatment of infants. It also resulted in showing that the late intensification of chemotherapy did not benefit patients [5]. The overall 5-year EFS rate was 46.1% and the survival was 55.2% for the whole study cohort. Large cohort of infants with a presence of *KMT2A* rearrangement showed that only patients with additional unfavorable prognostic features (age less than 6 month, either white blood cells (WBC) counts ≥ 300 g/L or poor response to steroids at day 8) appeared to benefit from allogenic HCT [31].

In another study, 99 infants treated with the Interfant-99 protocol were comprised to prognose the significance of minimal residual disease (MRD) in infants with acute lymphoblastic leukemia. MRD was analyzed by PCR technique analyzing different data: rearranged immunoglobulin genes, T-cell receptor genes, and *KMT2A* genes at various time points (TP) during therapy. Higher MRD levels at the end of induction and consolidation (TP2 and TP3) were significantly associated with lower disease-free survival. Further analysis of TP2 and TP3 allowed the determination of three patients’ groups with different research results, as follows: all MRD-high-risk patients (MRD levels > or = 10^(−4)^ at TP3; 26% of patients) relapsed, MRD-low-risk patients (MRD level < 10^(−4)^ at both TP2 and TP3) constituted 44% of patients and showed a relapse rate of only 13%, whereas the remaining patients (MRD-medium-risk patients; 30% of patients) had a relapse rate of 31%. This analysis proved that MRD is an important prognostic factor and its diagnostics has added value for recognition of risk groups in infant ALL and that MRD analysis can be useful in establishing treatment interventions in infant ALL as well [32].

Based on an Interfant-99 study, researchers performed another Interfant-06 trial. A total of 651 infants with ALL were enrolled and divided into low risk (non-rearranged *KMT2A*), high risk (presence of a *KMT2A*-rearrangement and age < 6 months at diagnosis, with a WBC count 300 × 10^9^/L or more at diagnosis or a poor prednisone response), and medium risk (all other patients with *KMT2A*-r ALL). Patients in the medium and high-risk groups were randomly assigned to receive the lymphoid course low-dose cytosine arabinoside [araC], 6-mercaptopurine, cyclophosphamide (IB) or experimental myeloid cycles, namely araC, daunorubicin, etoposide (ADE), and mitoxantrone, araC, etoposide (MAE). The principal aims were to assess early intensification with post-induction myeloid-type chemotherapy to improve outcomes and prevent early relapse and to compare the results with Interfant-99, where the late intensification was performed. The study also differs from Interfant-99 due to the removal of dexamethasone and vincristine during maintenance. The new treatment approach did not significantly improve outcomes for infant ALL compared with the lymphoid-type course IB. There was also no significant difference in the 6-year EFS when comparing Interfant-06 to Interfant-99 (46.1% in both trials) [33].

## 4. Novel Therapies

The challenges in treating ALL in infants are multifaceted. Delivering effective chemotherapy requires dose modification due to developmental factors impacting drug pharmacokinetics differently than other age groups. Treatment-related toxicities, such as perineal irritation, mucositis, and skin breakdown, are dose-limiting issues. These pose serious infection risks, requiring aggressive preventive measures like antibiotic prophylaxis and surveillance for viral or fungal infections. Infant ALL survivors face significant long-term complications, with growth failure being the most common. This emphasizes the need to explore innovative therapies to mitigate toxicities associated with traditional chemotherapy and hematopoietic stem cell transplantation [34,35].

### 4.1. Blinatumomab

Blinatumomab, a CD3-CD19 bispecific T-cell engager (BiTE), comprises two recombinant single-chain variable fragments. By connecting CD19-positive B cells to CD3-positive T cells, blinatumomab activates T cells, releasing granzymes and perforins, which triggers a signaling pathway. Ultimately, leukemic B cells undergo destruction through caspase activation and apoptosis [36,37,38,39].

Examining blinatumomab in children under 18 with relapsed/refractory B-cell ALL, an open-label study conducted a phase 1 dosage escalation followed by a phase 2 part with 6-week treatment cycles. The established recommended dosage of 5/15 µg/m^2^/d revealed notable outcomes, also among 10 infants (8 with MLL translocations). Within two cycles, 60% of them achieved complete remission, and 40% proceeded to allo-HCT in remission [40,41].

In a different set of clinical reports, blinatumomab was administered to children with MRD-positive ALL, serving as a transitional strategy before undergoing transplantation. This included two infants with ALL who successfully attained a complete MRD response but experienced a relapse after the transplant [42].

A retrospective analysis examined 11 infants diagnosed with B-ALL who were treated with blinatumomab for MRD reduction before allo-HCT. All patients had *KMT2A* rearrangement. The results showed a 100% partial or complete MRD response rate, with nine patients becoming MRD negative. The median time from commencing blinatumomab to HCT was 51 days. The 3-year EFS and OS post-transplant were 47% and 81%, respectively. Four patients relapsed post-transplant, and one patient with a myeloid lineage switch died of progressive leukemia [43].

The potential efficiency of blinatumomab therapy for infants with ALL, alongside the less-than-optimal outcomes observed in Interfant-06 trial (six-year OS 58.2%) [33], highlight an urgent need for new studies to investigate the possible benefits of a combined approach.

This has led to the initiation of a prospective, single-arm, phase 2 study recruiting infants with *KMT2A*-r ALL. Patients were treated according to the standard Interfant-06 protocol and those with post-induction M1/M2 marrow received one cycle of blinatumomab. The complete course of blinatumomab was administered to all 28 patients without any treatment interruptions. A complete response with negative MRD was observed in 54%, indicating a tendency towards a higher rate compared to the end of consolidation in the Interfant-06 study (40%). After a 26.3 months follow-up period, the two-year EFS was 81.6%, whereas the two-year OS reached 93%. According to Interfant-06, the same criteria were 49.4% and 65.8%, respectively. Despite the study’s limited patient population and a relatively short observation period, its outcomes can be deemed a significant success in the treatment of infants with ALL using blinatumomab [44]. These findings will be utilized in the Interfant-21 study, where patients will undergo two cycles of blinatumomab before HCT [45].

In another phase 3 clinical trial, infants with ALL are being categorized into three groups: low risk with no *KMT2A* rearrangements, intermediate risk *KMT2*-ALL without central nervous system damage, and high risk with central nervous system lesions. For high-risk patients, a combination of blinatumomab and allo-HCT in the first remission has been selected. No results have been posted yet [46].

### 4.2. CAR-T Cells

CAR-T cell therapy has demonstrated remarkable success in treating pediatric refractory/relapsed B-ALL, significantly improving outcomes for children who may not have responded well to conventional treatments [47]. Infants were initially excluded from studies involving CAR-T cell therapy due to concerns about the heightened risk of toxicity in this age group. Additionally, they have a low circulating blood volume, which poses a challenge for effective apheresis. Problems also include obtaining a sufficient number of T lymphocytes and the potential impairment of their function [48]. Furthermore, reports of lineage switch in *KMT2A*-r leukemia after CD19-targeted therapy raise concerns about an elevated risk of myeloid leukemia relapse following B-lineage CAR-T cell therapy in this group [49,50].

Quasim et al. created untypical, universal CAR19 (UCART19) cells by using lentiviral transduction of donor cells matched to non-human leukocyte antigens. They simultaneously edited the genes of the T cell receptor’s alpha chain and the CD52 gene loci using an effector nuclease-like transcription activator (TALEN). This bridging therapy was applied to two infants, allowing for a 28-day remission before a subsequent and successful HCT [51,52].

PLAT-02 and PLAT-05 are clinical trials investigating feasibility and efficiency of CD-19 and CD-22 specific CAR-T cells in infants with relapsed/refractory *KMT2A*-r B-ALL following lymphodepleting chemotherapy. The primary inclusion criterion was a lymphocyte absolute count exceeding ≥100 cells/µL. Among 18 infants, CAR-T cell products were successfully manufactured in 17/18 subjects. Of these, 16/17 were infused, with a median follow-up of 26.9 months. The highest CRS grade observed was 3, affecting 2 out of 15 assessable subjects (13%), while neurotoxicity was restricted to a maximum grade of 2. The majority achieved MRD-negative complete remission (93.3%) by day 21, and estimated the one-year LFS was 66.7%, while the one-year OS was 71.4% Outcomes are favorable due to the comparable toxicity and MRD-CR rates observed in non-infant ALL cases [53].

In another study, 13 infants treated with tisagenlecleucel achieved a two-year RFS of 67% and OS of 65%. Nevertheless, it has been shown to have a higher risk of relapses due to myeloid leukemia switch [54,55].

Results from a retrospective study of 14 infants suggest that patients who were in morphologic remission with or without MRD at the time of infusion were able to achieve and largely maintain MRD-free remission after CAR-T. In turn, those children who at the time of infusion were in an advanced stage of the disease (bone marrow > M1) did not respond to this therapy [56].

### 4.3. Menin Inhibitors

Menin inhibitors are designed to bind KMT2A pocket on menin, preventing the formation of the menin-KMT2A fusion protein complex leading to a rapid inhibition of its expression. This interaction is critical for the abnormal gene patterns observed in leukemia with *KMT2A* rearrangements [57,58].

Spectacular results of menin inhibitors were shown in mouse models with *KMT2A*-r ALL xenografts, wherein leukemia burden was dramatically reduced offering the potential for prolonged remission even with the use of a single agent [59]. This led to the development of a clinical trial aimed at assessing the efficacy of SNDX-5613. They are recruiting patients, including infants, with relapsed/refractory ALL, particularly those with *KMT2A* gene rearrangements, or *NPM1* mutations [60]. Another clinical study utilizing the SNDX-5613 is currently in development, specifically recruiting infants and children under the age of 6 [61].

### 4.4. BCL-2 Inhibitors

BCL-2 is a protein that plays a crucial role in regulating apoptosis processes. Studies have shown that *KMT2A*-r ALL blasts exhibit an overexpression of BCL-2, potentially leading to the heightened resistance of cancer cells against programmed death [62]. Therefore, targeting BCL-2 proves to be an appealing strategy, as evidenced by significant responses observed in infant patient-derived xenografts with *KMT2A* rearrangements following treatment with venetoclax [63,64].

Based on this evidence, a phase 1, open-label clinical trial was conducted to evaluate the safety and pharmacokinetics of venetoclax in monotherapy or combined with chemotherapy in pediatric and young adult patients. Among the 11 children with ALL enrolled in the study, there were also infants included. The ORR was 27% in both parts. The most common treatment emergent adverse events (TEAEs) were vomiting, diarrhea, hypokalemia, increased ALT, febrile neutropenia, and anemia. Notably, two patients achieved complete remission with incomplete marrow recovery (CRi) after adding 1 or 2 cycles of venetoclax in combination with dexamethasone-vincristine-peg-asparaginase. Additionally, three patients demonstrated MRD negativity, indicating a positive treatment outcome [65,66].

### 4.5. FLT3 Inihibitors

*FLT3* is a gene encoding a tyrosine kinase receptor that regulates the growth and differentiation of hematopoietic cells. In the context of ALL development, this results in uncontrolled proliferation and survival of leukemic cells. *FLT3* mutations are more typical for AML, but they can manifest in a specific subset of ALL cases [67]. While activating mutations in *FLT3* are relatively rare in infants suffering from ALL, the overexpression of wild-type *FLT3*, causing autoactivation, is a notable characteristic and high-risk factor in *KMT2A*-r leukemias [68].

A phase 3 clinical trial conducted by the Children’s Oncology Group recruited patients with ALL under 366 days of age. The influence of the integrating *FLT3* inhibitor lestaurtinib following previous post-induction chemotherapy on EFS was examined. However, the results showed no significant difference in the 3-year EFS between infants treated with chemotherapy plus lestaurtinib (*n* = 67.36 ± 6%) and those receiving chemotherapy alone (*n* = 54.39 ± 7%, *p* = 0.67). Subsequent pharmacodynamic assays proved that patients achieving strong FLT3 inhibition in plasma and those with leukemic cells sensitive to the ex vivo FLT3-inhibition assay achieved a better EFS. A total of 17 patients exhibited both inhibition and sensitivity, resulting in a 3-year EFS of 88 ± 8%. Overall, the study showed that along with optimal patient selection, lestaurtinib added to chemotherapy led to favorable outcomes in the subgroup of infants suffering from *KMT2A*-r ALL [69].

### 4.6. Nucleoside Analogues

Clofarabine as a second-generation, purine nucleoside analogue is responsible for orchestrated DNA disruption processes and the promotion of apoptosis in neoplasm cells [70]. A preclinical study investigating the efficiency of nucleoside analogues in infantile, *KMT2A*-r ALL line cells demonstrated the highest activity of clofarabine. Synergistic cytotoxicity in combination with cytarabine has also been proven. Moreover, clofarabine induced the demethylation of the promoter region in the tumor suppressor gene *FHIT* (Fragile Histidine Triad), which is typically hypermethylated in infant *KMT2A*-r ALL [71].

Subsequently, among the 12 infants with ALL enrolled in the phase 3 clinical study, 9 had the *KMT2A-r* subtype and received a regimen containing clofarabine. The 5-year EFS and OS were estimated at 44.4% and 55.6%, respectively. Six infants initially tested positive for MRD, and four of them attained MRD negativity, while the remaining two exhibited a reduction in MRD levels. At the time of publication, five patients stayed alive and four died due to infection (three) and pulmonary hypertension (one). The results of the study showed that while clofarabine demonstrated efficacy in reducing MRD in infants with *KMT2A*-r ALL, caution in its upfront use due to significant treatment-related mortality, primarily attributed to grade 5 infections. It suggests contemplating clofarabine as a salvage alternative in case other immunotherapeutic strategies prove ineffective [72]. Novel approaches for infant ALL treatment are visualized in Figure 3. Next generation clinical trials based on new therapeutic strategies are summarized in Table 2.

### 4.7. Epigenetic Modifiers

*KMT2A*-r ALL infant cells exhibit abnormal DNA methylation patterns, as revealed by genome-wide methylation studies. Studies suggest that leukemia-specific histone modifications, like H3K79 dimethylation induced by a disruptor of telomeric silencing 1-Like (DOT1L) recruitment in KMT2A fusion proteins, can be effectively controlled by histone deacetylase (HDAC) inhibitors. These findings support the potential use of demethylating agents or HDAC inhibitors to reverse inherent chemotherapy resistance in infant *KMT2A*-r ALL [77,78,79].

COG created a phase 2, single-arm, pilot trial to assess the safety of incorporating azacitidine into the standard Interfant-06 protocol for infants newly diagnosed with *KMT2A*-r ALL. A total of 53 infants treated with chemotherapy and at least one course of azacitidine, with a median follow-up 3.8 years, achieved EFS and OS rates of 34.2% and 63.8%, respectively. Treatment failure was observed in six infants. It was proven that EFS significantly correlates with MRD. EFS among infants with any positive MRD was 20.6%, contrasting with the 40.1% EFS for those without MRD. However, the results are not satisfactory, highlighting the need for improvement and further research in new therapies for infants suffering from ALL [74].

The TINI studies are currently testing the feasibility of adding bortezomib (a proteasome inhibitor), and vorinostat (a histone deacetylase inhibitor) to the chemotherapy regimen in infants newly diagnosed with ALL [75,76].

## 5. Summary

Treatment of infant ALL has faced numerous hurdles. In recent decades, intensification of chemotherapy has improved overall survival outcomes in infant patients with ALL, whereas inferior therapeutic results were strongly related with the *KMT2A*-r subtype. Furthermore, future studies should consider managing with ALL treatment complications to maximize efficacy and minimize toxicity of the therapy. Better understanding of the infant ALL and worldwide development of immunotherapy led to an increased number of novel treatment options and the next generation of promising clinical trials. Infant ALL still remains a challenge; hence, it is pivotal to continue discovery of new targets and treatment strategies.

## Figures and Tables

**Figure 1 ijms-25-03721-f001:**
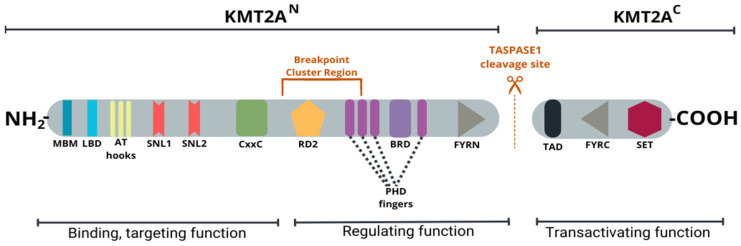
Wild-type *KMT2A* domain structure. Proteolytic cleavage of the KMT2A molecule by enzyme TASPASE1 at the cleavage site (indicated by the orange dashed line) results in the formation of two subunits, bigger KMT2AN and smaller KMT2AC. The KMT2AN subunit includes menin-binding motif (MBM), lens epithelium-derived growth factor (LEDGF)-binding domain (LBD), adenosine–thymidine hook (AT hooks), which nonspecifically bind the minor groove of DNA, CxxC (cysteine-rich type of zinc finger domain) region-specific for non-methylated DNA, subnuclear localization domains (SNL1, SNL 2), repression domain 2 (RD2), four plant homeodomain (PHD) fingers, bromodomain (BRD) and F- phenylalanine, Y-tyrosine rich N-terminal (FYRN) domain. The KMT2AC subunit contains transactivation domain (TAD), a FY-rich C-terminal (FYRC) domain and C-terminal SET (Su(var)3-9, Enhancer-of-zest e and Trithorax) domain. CxxC and RD2 regions have intrinsic activity in transcriptional repression. PHD and BRD are essential for the post-translational regulation and mediation of protein–protein interactions. FYRN and FYRC are necessary for interactions between the KMT2AN and KMT2AC subunits and subnuclear localization of this complex. The SET domain is required for KMT2A’s H3K4 methyltransferase activity. The breakpoint cluster region spans exons 9–14 and the KMT2A’s fusion breakpoints, where fusion partners are attached, are typically located within this area. The image was created with Canva Pro https://www.canva.com/pro/ (accessed on 29 January 2024).

**Figure 2 ijms-25-03721-f002:**
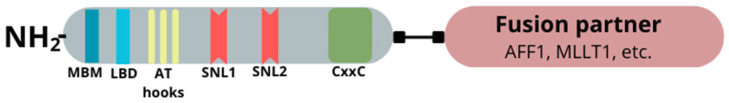
Structure of a KMT2A fusion protein. MBM—menin-binding motif; LBD—LEDGF-binding domain; AT hooks—adenosine–thymidine hook; SNL1, SNL2—subnuclear localization domains; CxxC—cysteine-rich type of zinc finger domain. Image created with Canva Pro https://www.canva.com/pro/ (accessed on 29 January 2024).

**Figure 3 ijms-25-03721-f003:**
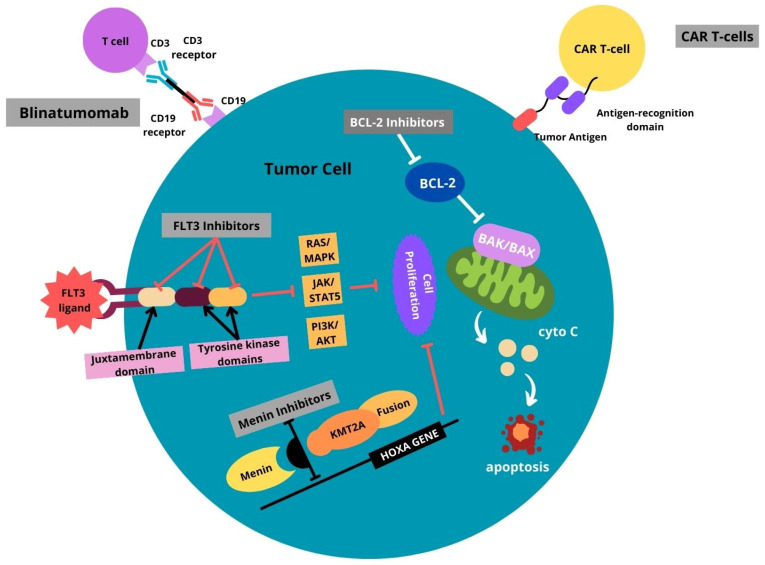
Summary of novel approaches for infant ALL treatment. FLT3 ligand, Fms-related tyrosine kinase 3 ligand; FLT3 inhibitors, Fms-related tyrosine kinase 3 inhibitors, RAS/MAPK, Ras/mitogen-activated protein kinase; JAK/STAT5, Janus kinase/signal transducer and activator of transcription 5; PI3K/AKT, Phosphoinositide 3-kinases/Serine/threonine kinase; KMT2A, histone-lysine N-methyltransferase 2A; BCL-2, B-cell lymphoma 2; BCL-2 inhibitors, B-cell lymphoma 2. Image created with Canva Pro https://www.canva.com/pro/ (accessed on 29 January 2024).

**Table 1 ijms-25-03721-t001:** *KMT2A* fusion partners.

*KMT2A* Fusion Partner	Former Terminology	Frequency in Infant ALL	Type and Localization of Abnormality
*AFF1*	*AF4*	48%	t(4;11) (q21;q23)
*MLLT1*	*ENL*	24%	t(11;19) (q23;p13.3)
*MLLT3*	*AF9*	16%	t(9;11) (p21;q23)
*MLLT10*	*AF10*	6%	t(10;11) (p12;q23)
Other	-	6%	-

**Table 2 ijms-25-03721-t002:** Novel clinical trials for infant ALL treatment.

Drug Class	Drug	ClinicalTrials.Gov Identifier	Phase	Completed/Ongoing	Responsible Party	Key Results	References
BiTE	Blinatumomab	NCT01471782	1/2	Completed	Amgen Research (Munich) GmbH	OS 40% at the 24-month follow-up	[40,41]
		NCT05327894 (Interfant-21)	3	Ongoing	Princess Maxima Center for Pediatric Oncology	N/A	[45]
		NCT05029531	3	Ongoing	Federal Research Institute of Pediatric Hematology, Oncology and Immunology	N/A	[46]
CAR-T cells	CTL019	NCT02435849	2	Completed	Novartis Pharmaceuticals	1-year EFS 50%	[48]
	UCART19	NCT02808442	1	Completed	Institut de Recherches Internationales Servier	MRD negativity (2 pts) at the 18-month follow-up	[52]
	CD19-CAR T Cells also expressing an EGFRt	NCT02028455	1/2	Ongoing	Seattle Children’s Hospital	1-year OS 71.4%	[53]
	CD19- and CD22 specific CAR-T cells	NCT03330691	1	Ongoing	Seattle Children’s Hospital	N/A	[73]
Menin Inhibitors	Revumenib (SNDX-5613)	NCT04065399	1/2	Ongoing	Syndax Pharmaceuticals	N/A	[60]
	Revumenib in	NCT05761171	2	Ongoing	Children’s Oncology Group	N/A	[61]
BCL-2 inhibitors	Venetoclax	NCT03236857	1	Completed	AbbVie	ORR 27%; MRD negativity (3 pts)	[65]
FLT3 Inhibitors	Lestaurtinib	NCT00557193	3	Ongoing	Children’s Oncology Group	3-year EFS 88 ± 8%	[66,69]
Nucleoside analogues	Clofarabine	NCT00549848	3	Completed	St. Jude Children’s Research Hospital	5-year EFS 44.4%; OS 55.6%	[72]
Epigenetic modifiers	Azacitidine	NCT02828358	3	Ongoing	National Cancer Institute	EFS 34.2%; OS 63.8% with median follow-up 3.8 years	[74]
	Bortezomib and vorinostat	NCT02553460	1/2	Ongoing	St. Jude Children’s Research Hospital	N/A	[75]
-	bortezomib, vorinostat, blinatumomab, ziftomenib	NCT05848687	1/2	Ongoing	Tanja Andrea Gruber, Stanford University	3-year EFS 56.5%; OS 70.5%	[76]

OS—overall survival; EFS—event free survival; pts—patients; MRD—minimal residual disease; ORR—objective response rate; N/A—not available.

## Data Availability

Not applicable.

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
