# Peer review of "Infant Acute Lymphoblastic Leukemia—New Therapeutic Opportunities"

_ijms, 2024, doi:10.3390/ijms25073721_

Round 1

Reviewer 1 Report

Comments and Suggestions for Authors
Kulczycka and colleagues review the current treatment options for Infant ALL. A detailed description of the protocols carried by the main collaborative groups is presented, and current clinical trials testing state-of-the art therapies are described. The manuscript is comprehensive and overall well written. However, some points should be improved prior to publication. 

- I understand that KMT2A germline refers to wild type (non-rearranged) patients. Although this nomenclature is used in some papers, I suggest use the term non-rearranged or wild type KMT2A, which is more commonly used in the literature.

- Some parts of the introduction are misleading. For example, in Line 38, it should be clear that  KMT2A-r is not equivalent to MLL-mixed lineage leukemia. Mixed lineage leukemia is characterized by the presence of MLL fusion and high risk features, but KMT2A-r cam be algo present in other acute leukemias. 

- Please, use commonly accepted vocabulary (forecast, line 32; weak, line 40; unsatisfactory outcomens, line 155). Although the English language is generally well used, review the  whole manuscript . 

- Use consistent nomenclature along the manuscript: KMT2A-ALL; KMT2A-r; CAR T, CAR-T

- Table 2 would be more informative if survival or main conclusions from completed trials were included. 

Comments on the Quality of English Language

Please, review grammar and use the most appropriate vocabulary

Reviewer 2 Report

Comments and Suggestions for Authors

A well-written and interesting manuscript on ALL in infants. The topic is not discussed much in more broad-based journals such as IJMS, so it will certainly be of interest to readers. The senior author of this publication is also a well-known clinical hematologist, with an established scientific position, especially in the field of AL and CAR-T.

Below are my comments on the manuscript:

1. in general, the content is detailed and well-written, and the overall impression is very good, but it would be worth having the paper analyzed by a native speaker,

2. it would be good for authors to provide institutional e-mail addresses, even if they are students,

3. HSCT or is it better HCT?,

4. Fig. 3 could be of better quality,

5. please expand the summary with future perspectives, e.g. what the treatment of complications of ALL therapy may look like, e.g. resulting from frequent transfusions, such as iron overload (Cancers (Basel). 2023;15(4):1041; Children (Basel). 2023 ;10(5):870.)

Comments on the Quality of English Language

Minor editing of the English language is required.

Round 2

Reviewer 2 Report

Comments and Suggestions for Authors

I think that the authors have adequately addressed the comments made by the reviewers.